# Retrieval of stratospheric aerosol size distribution parameters using SAGE-III/ISS extinction measurements at three wavelengths

Felix Wrana[1], Christian von Savigny[1], Jacob Zalach[1], and Larry W. Thomason[2]

[1]Institute of Physics, University of Greifswald, Felix-Hausdorff-Str. 6, 17489 Greifswald, Germany
[2]NASA Langley Research Center, Hampton, Virginia, USA

**Correspondence:** Felix Wrana
(felix.wrana@uni-greifswald.de)

**Abstract.**

In this work a novel approach to determine the particle size distribution (PSD) parameters of stratospheric sulfate aerosols is presented. For this, ratios of extinction coefficients obtained from SAGE III/ISS solar occultation measurements at 449 nm, 756 nm and 1544 nm were used to retrieve the mode width and median radius of a size distribution assumed to be monomodal lognormal. The estimated errors at the peak of the stratospheric aerosol layer on average lie between 20 % and 25 % for the median radius and 5 % and 7 % for the mode width. The results are consistent in magnitude with other retrieval results from the literature, but a robust comparison is difficult, mainly because of differences in temporal and spatial coverage. Other quantities like number density and effective radius were also calculated. A major advantage of the described over other retrieval techniques is that both the median radius and the mode width can be retrieved simultaneously, without having to assume one of them. **This is possible due to the broad wavelength spectrum covered by the SAGE III/ISS measurements. Also, the presented method – being based on the analysis of three wavelengths – allows unique solutions for the retrieval of PSD parameters for almost all of the observed extinction spectra, which is not the case when using only 2 spectral channels. In addition**, the extinction coefficients from SAGE III/ISS solar occultation measurements, on which the retrieval is based, are calculated without a priori assumptions about the PSD. For those reasons, the data produced with the presented retrieval technique may be a valuable contribution to better understand the variability of stratospheric aerosol size distributions, e.g. after volcanic eruptions. While this study focuses on describing the retrieval method and a future study will discuss the PSD parameter data set produced in depth, some exemplary results for background conditions in June 2017 are shown.

## 1 Introduction

The existence of a permanent aerosol layer in the stratosphere, typically known as the "Junge-layer" is known already since the late 1950s, when Christian Junge performed balloon-borne in situ measurements there (Junge et al., 1961). The layer resides roughly between 15 and 30 km in the lower stratosphere directly above the tropopause. The aerosols are usually assumed to be droplets of a solution of sulfuric acid and water with a weight percentage of sulfuric acid of around 75% (Rosen, 1971; Arnold

et al., 1998), although small but still relatively uncertain contributions of other compounds such as carbonaceous and meteoric material are possible (Murphy et al., 2007).

Anthropogenic $SO_2$ emissions play a role in the variation of the stratospheric aerosol (SA) budget (Sheng et al., 2014), but they are dwarfed by natural sources, especially by direct injections due to large volcanic eruptions. **Intense biomass burning events, such as the Canadian wildfires in 2017 (Ansmann et al., 2018) and the Australian bushfires of 2019/2020 (Ohneiser et al., 2020), can also play an important role in the feeding of the aerosol layer.** Nevertheless, the Junge layer is persistent globally over time, even in volcanically quiescent periods **and without large biomass burning events** (Kremser et al., 2016). The stratospheric aerosol layer is then mainly sustained by a flux of sulfurous aerosols and precursor gases from the troposphere, such as $SO_2$ and OCS (Sheng et al., 2014), which are eventually oxidized to $H_2SO_4$, which itself forms new $H_2SO_4$-$H_2O$-droplets by co-condensation with water vapor (Hamill et al., 1990). The formed aerosols grow through coagulation and further condensation, while sedimentation limits the **averaged** aerosol size in the aerosol layer. **In the stratosphere,** evaporation due to rising temperatures with height generally determines the upper boundary of the aerosol layer. The most important region for this transport of sulfur bearing substances from the troposphere to the stratosphere is the tropical tropopause layer (TTL) (Kremser et al., 2016).

Stratospheric aerosols play a role in the chemistry of Earth's atmosphere as well as its radiative balance. Regarding the former, they influence the levels of different atmospheric constituents, like $NO_x$ (Deshler, 2008) and stratospheric ozone, when SA levels are elevated due to volcanic eruptions (Hofmann & Solomon, 1989; Gleason et al., 1993). Also they can act as condensation nuclei for the formation of polar stratospheric clouds (PSCs), which are central in catalytic ozone destruction during polar winter and spring (Deshler, 2008).

Concerning the radiative balance of the atmosphere, SA absorb and emit longwave radiation, thereby having a warming effect on the stratosphere, and contribute to the extinction of solar radiation mainly by scattering, thus leading to a cooling of the troposphere (Dutton & Christy, 1992). All of those effects are critically dependent on the size distribution of the aerosols, with smaller particles at a constant aerosol mass being more efficient at destroying ozone (Robock, 2015) and larger particles being more efficient at scattering solar radiation, which leads to the aforementioned cooling of the troposphere. However, the cooling effect is dominant only up to a certain size (effective radius of about 2 microns), above which their absorptive capacity surpasses their scattering one and they have a net warming effect on Earth's surface (Lacis et al., 1992).

The great importance of the size of stratospheric aerosols is the reason, why Robock (2015) stated the question about changing aerosol size after large $SO_2$ injections into the stratosphere to be one of the most outstanding research questions regarding the link between volcanic eruptions and the associated climate response.

Mathematically, the size distributions of SA are usually expressed as lognormal functions (see Eq. (1)), based on fits to the longest available set of in situ measurements carried out from Laramie, Wyoming (Deshler et al., 2003). A monomodal lognormal distribution is expressed as follows:

$$\frac{dN(r)}{dr} = \frac{N_0}{\sqrt{2\pi} \cdot r \cdot ln\sigma} \cdot exp(-\frac{ln^2(r/r_{med})}{2ln^2\sigma}) \tag{1}$$

where $r_{med}$ is the median radius, $\sigma$ the mode width of the lognormal distribution and $N_0$ the total number density. The scattering and absorption cross sections, and thereby the extinction cross section of SA can be calculated using Mie theory as a function of their composition and the **particle size distribution (PSD)** parameters (Mie, 1908). Therefore, those parameters can in principle be retrieved from extinction coefficients calculated from measurements at multiple wavelengths, with the particle composition in this case being a secondary factor, since the realistic range of the real refractive indices of $H_2SO_4$-$H_2O$ aerosol droplets is not very large. As a source of these extinction coefficients, satellite data is particularly valuable as, depending on the orbit parameters, a near global coverage is possible.

The aim of this work is to retrieve the stratospheric aerosol size distribution parameters from the extinction measurements of the SAGE III instrument mounted on the International Space Station (Cisewski et al., 2014). Here we assume the lognormal size distribution to be monomodal, which is a common assumption. Please note that we do not claim that the actual stratospheric aerosol size distribution is well described by a monomodal lognormal distribution under all circumstances. While Deshler et al. (2003) regularly fit a bimodal lognormal distribution to their in situ measurements, the existence of a second mode is still controversial, with a gamma distribution also being discussed **(Wang et al., 1989; Nyaku et al., 2020)**. Also, due to the limited degrees of freedom, to retrieve the parameters of a bimodal lognormal distribution with the retrieval technique presented in the current work would not be possible. While there are studies in which both the median radius and the mode width of a monomodal lognormal PSD have been retrieved **(Bingen et al., 2002; Wurl et al., 2010; Malinina et al., 2018)**, with other data sets in the past it was often necessary to fix either the median radius or the mode width to determine the other (Yue and Deepak, 1983; Bourassa et al., 2008; Zalach et al., 2020) **or it was only possible to retrieve a range of plausible mode width values (Bauman et al., 2003)**. In this work both parameters are retrieved simultaneously, which is possible because of the broad wavelength spectrum of the SAGE III/ISS instrument and the retrieval method described in this study. This method was used successfully in the past to retrieve the PSDs of noctilucent cloud (NLC) particles, or polar mesospheric cloud (PMC) particles (von Cossart et al., 1999; Baumgarten et al., 2006).

This work is part of the cooperative research project "VolImpact" (Revisiting the volcanic impact on atmosphere and climate – preparations for the next big volcanic eruption), which focuses on the response of the climate system to volcanic eruptions (von Savigny et al., 2020). In particular it is part of the project VolARC (Constraining the effects of Volcanic Aerosol on Radiative forcing and stratospheric Composition). One of the foci of this project is to better understand the temporal variability of the PSD of stratospheric aerosols.

The main purpose of the present study is to introduce the PSD retrieval approach applied. A full analysis and discussion of all the results obtained from the SAGE III/ISS solar occultation measurements will be the topic of a future publication. After shortly introducing the data set used in Sect. 2, we will present the method with which the size distribution parameters were retrieved in Sect. 3 and explain the error calculations made in Sect. 4. Afterwards, latitudinal contour plots and sample vertical profiles of the most important parameters are shown and discussed in Sect. 5, with conclusions in Sect. 6.

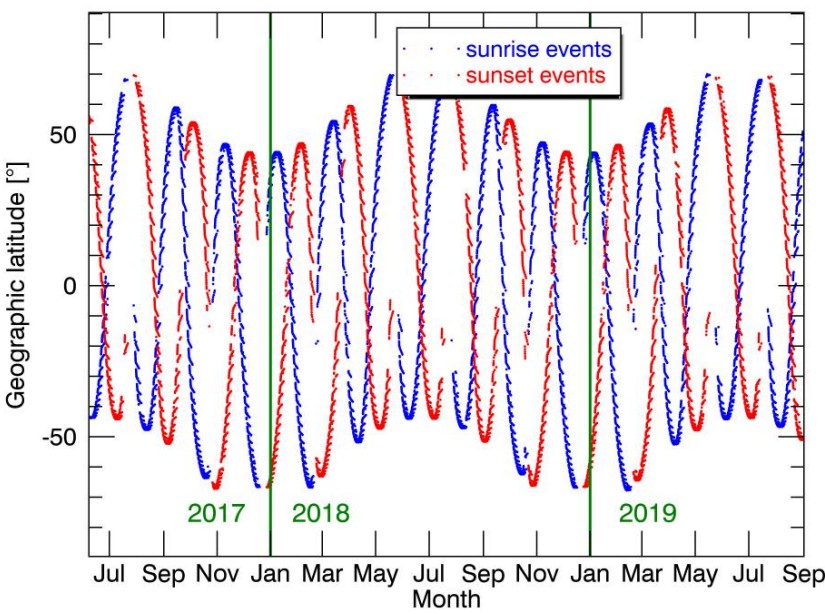

**Figure 1.** Latitudinal coverage of SAGE III/ISS solar occultation measurements between June 2017 and September 2019. Observed sunrise and sunset events are shown in blue and red, respectively. Green vertical lines mark turns of the year.

## 2 SAGE III/ISS instrument

The satellite data set that was used in this work to retrieve the PSD parameters comes from the SAGE III instrument (Stratospheric Aerosol and Gas Experiment) that is mounted on the International Space Station (ISS) since 2017 and is a part of NASA's "Earth Observing System". The instrument is the successor of the satellite experiments SAM II, SAGE I, SAGE II and SAGE III Meteor-3M and performs lunar and solar occultation measurements, measuring the attenuation of solar radiation due to scattering and absorption by atmospheric constituents such as ozone, water vapor and aerosols. On board the ISS it ob-

serves around 15 sunrise and 15 sunset events in 24 hours, respectively. While occultation measurements are characterized by a limited spatial and temporal coverage, an important advantage as opposed to limb-scatter measurements is that atmospheric extinction can be obtained directly without a priori knowledge of the PSD and phase function. The SAGE III/ISS level 2 solar aerosol product used (version 5.1) contains aerosol extinction coefficients from tangent heights 0 to 45 km with a grid step size of 0.5 km and a maximum latitudinal range roughly between 69° N and 67° S. The latitude of the sunrise and sunset

measurements oscillates with a period of about 2 months, which can be seen in Figure 1, where geographic latitude of each event is plotted as a function of time for the available data from June 07, 2017 up until April 30, 2019 (NASA, 2019).

**Table 1.** Spectral channels of the SAGE III/ISS level 2 solar aerosol product and the respective extinction measurement uncertainties, as provided by NASA Atmospheric Science Data Center (NASA, 2019), averaged from June 2017 to December 2019 at 20 km altitude.

| $\lambda$ [nm] | Relative uncertainty |
|---|---|
| 384.224 | 0.0526 |
| 448.511 | 0.0399 |
| 520.513 | 0.0566 |
| 601.583 | 0.1589 |
| 676.037 | 0.0907 |
| 755.979 | 0.0319 |
| 869.178 | 0.0397 |
| 1021.20 | 0.0453 |
| 1543.92 | 0.0878 |

The aerosol extinction coefficients are available at 9 wavelengths, shown in Table 1. The 809 pixel CCD array used for the first 8 channels measures solar radiance with 1 to 2 nm spectral resolution between 280 and 1040 nm, while the 1543.92 nm channel data is based on measurements with an InGaAs infrared photodiode at 1550 nm with a 30 nm bandwidth. Also the relative uncertainties of the extinction measurements of each channel averaged between June of 2017 and April of 2019 for the altitude of 20 km are shown in Table 1. Especially the 1543.92 nm near infrared channel significantly extends the spectral range of extinction measurements when compared to SAGE II, the predecessor of this instrument, improving the precision of the method used in this work and rendering a simultaneous retrieval of both the median radius and mode width possible, as will be shown here.

## 3   Methodology

In this work a method similar to the one, that von Cossart et al. (1999) used to retrieve the size distribution parameters of noctilucent cloud (NLC) particles from lidar measurements, was implemented to derive the median radius and mode width of the stratospheric aerosol size distributions from the SAGE III/ISS aerosol extinction coefficients. For this, at each measured tangent height of a sunrise or sunset event two ratios of extinction coefficients at three wavelengths from SAGE III/ISS were compared to extinction ratios calculated with a Mie code provided by Oxford University's Department of Physics (Oxford, 2018) with predefined PSD parameters at the same wavelengths. These Mie calculations form the basis of the set of curves shown in the left panel of Figure 2 (discussed below), which is the main tool used for the retrieval and functions as a lookup table.

A number of assumptions were made for the Mie calculations and for the retrieval method to be applicable. Sensitivity testing of the retrieval to some of these parameters is discussed in chapter 4.

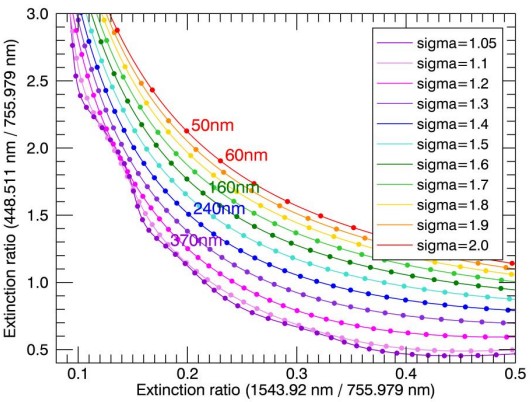 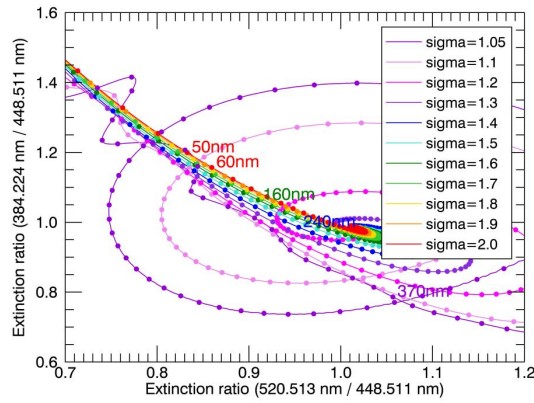

**Figure 2.** Extinction ratios at three wavelengths as a result of Mie calculations for values of median radii between 1 and 1000 nm and mode widths between 1.05 and 2.0. The dots mark $r_{med}$ values with increments of 10 nm. **The respective $r_{med}$ values of five selected dots are shown in the color corresponding to their mode width.** Left panel shows the set of curves to be compared to SAGE III/ISS measurements for the PSD parameter retrieval. Right panel shows an unusable set of curves with broad areas of non-unique solutions for the PSD parameters due to the closeness of the wavelengths used.

- The particles are assumed to consist of a solution of 75 % $H_2SO_4$ and 25 % $H_2O$ by weight, without other components such as meteoric or carbonaceous material. **As a result of this composition and due to being liquid droplets in the sub-micron size range, the particles can also be assumed to be spherical. Because of this, Mie theory can be applied. This composition also determines the wavelength-dependent real part of the refractive index $n$, that is used in Mie calculations.**

- The imaginary part of the refractive index $k$, or the absorption index is set to zero, since absorption for SA is very low for visible and near infrared radiation (Palmer and Williams, 1975).

- A monomodal lognormal particle size distribution is used (see Eq. (1)). Therefore the retrieval of $r_{med}$, $\sigma$ and $N$ is the main objective of this paper.

- The physically reasonable range for the mode width of a SA size distribution is assumed to be 1.05 to 2.0 (see discussion below).

Figure 2 shows two sets of curves resulting from the Mie calculations at three wavelengths each. The left panel depicts the set of curves used for the PSD parameter retrieval in this work and the right panel shows an example with a "bad" choice of wavelengths, as will be discussed further below. Each colored curve in both panels consists of the extinction ratios calculated for one constant mode width and median radii between 1 nm and 1000 nm, in 1 nm increments. The dots on each curve are 10 nm apart, for visual clarity.

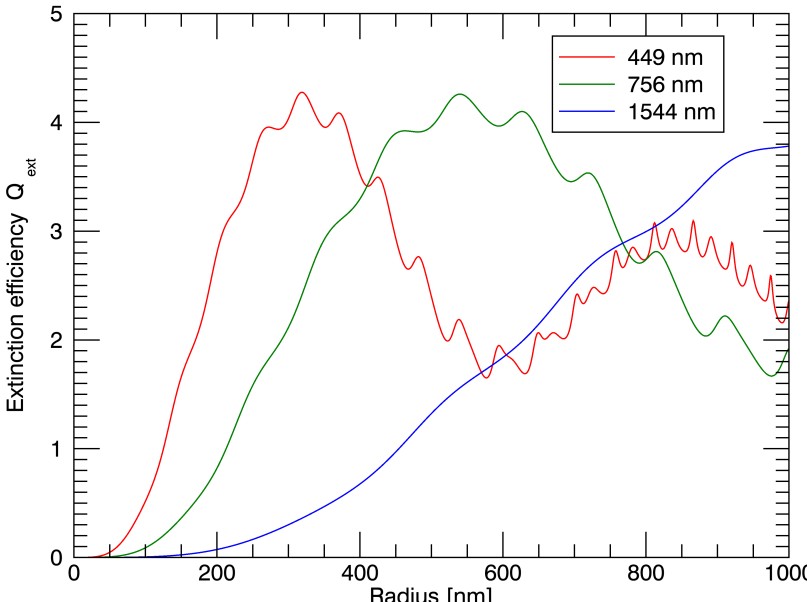

**Figure 3.** Extinction efficiencies of single (monodisperse) aerosols at radii between 1 nm and 1000 nm from Mie calculations for the three wavelengths used in the retrieval of the current work.

To obtain a total aerosol extinction coefficient $k_{ext}$ for a monomodal lognormal size distribution at a specific wavelength $\lambda$, median radius $r_{med}$ and mode width $\sigma$, single aerosol extinction coefficients calculated with the Mie Code (Oxford, 2018) are integrated over the radius range covered by the particle size distribution (PSD), as shown in Eq. (2). For the calculations the median radius and mode width of the size distribution, the total number density $N_0$ and the real and imaginary refractive index $n$ and $k$, which are determined by the assumptions made above, have to be assumed. For the calculation of extinction ratios, which are formed from these extinction coefficients and used for the actual size retrieval, the assumption about the number density is irrelevant though, since the extinction ratios become independent of $N_0$. The equation for the extinction coefficient looks as follows:

$$k_{ext}(\lambda) = \int\limits_0^\infty Q_{ext}(r, n, k, \lambda) \cdot \pi r^2 \cdot PSD(r, r_{med}, \sigma, N_0) \, dr \qquad (2)$$

Here, $Q_{ext}$ is the extinction efficiency of the single aerosol and $\pi r^2$ is the cross-sectional area of the spherical particle. Together, both quantities form the extinction cross section of the single aerosol particle.

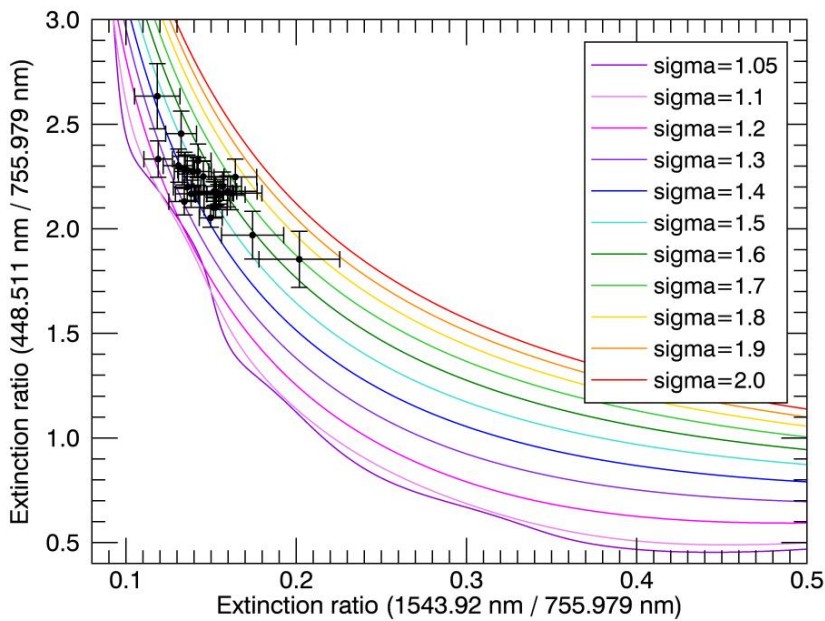

**Figure 4.** Extinction ratios derived from SAGE III/ISS measurements from one exemplary sunset event at $5.2°$ S and $179.6°$ W on June 23, 2017 plotted in the field of the same Mie calculations shown in the left panel of Figure 2, including error bars. Each point represents measurements at a different tangent height.

In Figure 3 the dependence of the extinction efficiency of a single $H_2SO_4$-$H_2O$ aerosol droplet on radius and wavelength is depicted for a radius range of 1 to 1000 $nm$ and the wavelengths 449 nm, 756 nm and 1544 nm, which are used in the PSD parameter retrieval of the current work. Since absorption is assumed to be zero, the extinction efficiency here is equal to the scattering efficiency. The spectral differences in the extinction efficiencies depending on the sizes of the aerosol in the size distribution are the basis for the retrieval of the unknown parameters of that distribution.

The values for the wavelength-dependent real refractive indices used in the Mie calculations to form the lookup table are based on the measurements performed by **Palmer and Williams (1975)** at 300 $K$. Lorentz-Lorenz-corrections, as described by **Steele and Hamill (1981)**, were conducted to obtain values at 215 K, which is typical for lower stratospheric temperatures. The temperature dependent density values of the $H_2O$-$H_2SO_4$ solution needed for the corrections were extrapolated from values from **Timmermans (1960)**.

With the lookup table ready, aerosol extinction coefficients from the SAGE III/ISS data set at the same three wavelengths are used to form extinction ratios, which can then be plotted into the 2-D space of the lookup table. This is shown in Figure 4, where the same set of curves is shown. Each dot with error bars represents a measurement of the same sunset event on June 23, 2017 at $5.2°$ S and $179.6°$ W at a different tangent height. Measurements which do not

fall within the space of the set of curves (around 4.7 % of measurements at 20 km), which is the case for noisy data, are not shown. Afterwards, $r_{med}$ and $\sigma$ can be derived for each tangent height by interpolating between the known values of two points of the Mie calculations above and below on the surrounding curves of the lookup table. This interpolation is done at the coordinates of the measurement data point, i.e. the error bars play no role in the retrieval itself, only in the identification and exclusion of noisy data later and in the choice of the wavelength combination, as will be discussed next.

The choice of the combination of the three wavelengths used for the retrieval is important, because the dependence of the extinction ratio on particle size is strongly dependent on $\lambda$, which in turn determines the shape of the calculated curves. For some wavelength combinations this leads to large areas in the 2-D space of extinction ratios, where sets of extinction ratios have multiple solutions for median radius and mode width (see right panel of Figure 2), while also leading to larger uncertainties in the results that can be retrieved. This is especially the case in $\lambda$ combinations that span only a small wavelength range, like 384.224 nm / 448.511 nm / 520.513 nm. In addition, the uncertainties of the extinction measurements performed by SAGE III/ISS also depend on the wavelength channel used, as can be seen in Table 1, in part due to differing contribution of Rayleigh scattering and sensitivity to aerosol. For the retrieval of this work, the $\lambda$ combination 448.511 nm / 755.979 nm / 1543.92 nm (left panel of Figure 2) was chosen, because it utilizes most of the wavelength range the SAGE III instrument offers, while avoiding potential problems with a systematic bias in the extinction coefficients in the 384.224 nm channel in the upper troposphere lower stratosphere (UTLS) region because of large contributions of molecular scattering there. Also, the averaged relative measurement uncertainty in the chosen spectral channels is as low as possible and the 601.583 nm and 676.037 nm channels, which have the highest uncertainties, are avoided. Although it has higher averaged relative extinction coefficient uncertainties, the 1543.92 nm channel is used instead of the one at 1021.2 nm. This is because the precision with which the median radius and mode width of the aerosol size distribution can be determined in the retrieval does not only depend on the uncertainty of the extinction ratios, i.e. their error bars, but also on how far the individual curves of the lookup table are apart (see Figure). Utilizing a much broader wavelength interval with the use of the 1543.92 nm channel increases this distance between the individual curves of the lookup table, overcompensating the higher extinction ratio uncertainties. This can be illustrated using an accuracy parameter that takes both factors into account and is used to assess the reliability of the $r_{med}$ and $\sigma$ values retrieved. This accuracy parameter $a$ for the retrieved PSD parameters at a specific tangent height is calculated as follows:

$$a = \frac{\Delta_x}{\delta_{f_x}} \cdot \frac{\Delta_y}{\delta_{f_y}} \qquad (3)$$

Here, $\Delta_x$ and $\Delta_y$ are the distances between the curves of the Mie calculations with the lowest and highest $\sigma$ in the direction of one axis, respectively, while $\delta_{f_x}$ and $\delta_{f_y}$ are the associated propagated measurement uncertainties of the extinction ratios. Therefore, small measurement uncertainties of the extinction coefficients at the three wavelengths

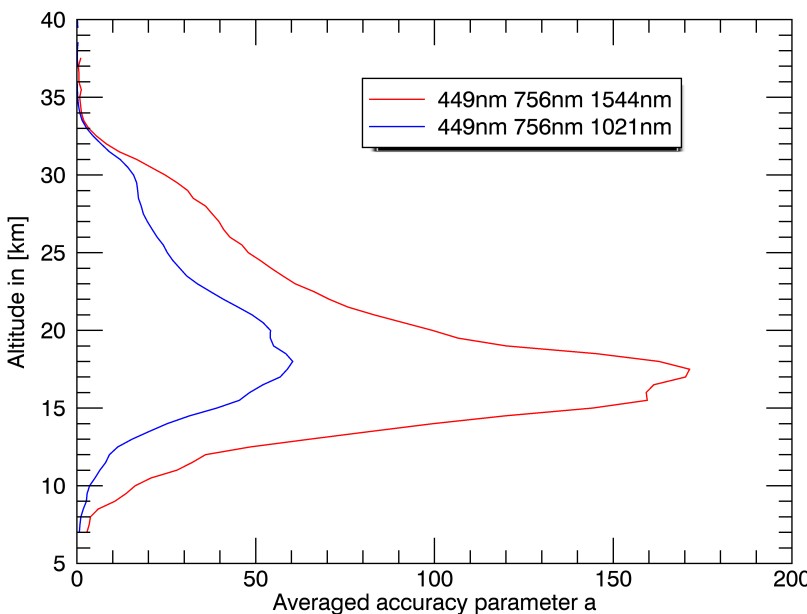

**Figure 5.** Profiles of the accuracy parameter $a$ averaged over the first 3000 solar occultation events of SAGE III/ISS for the wavelength combination 449 nm / 756 nm / 1544 nm (red) and 449 nm / 756 nm / 1021 nm (blue).

coupled with the curves of the Mie calculations being far apart lead to high accuracy parameters, indicating reliable
retrieval results.

In Figure 5 two profiles of the accuracy parameter $a$ averaged over the first 3000 solar occultation events observed by SAGE III/ISS are shown. The red curve is the profile using the wavelength combination 449 nm / 756 nm / 1544 nm, which is used in the retrieval of the PSD parameters, while the blue curve was calculated using the 1021.2 nm channel instead of the 1543.92 nm channel. The Plot clearly shows higher accuracy parameter values using the 1543.92 nm channel across all altitudes for the reasons discussed above, making this channel much better suited for the retrieval method presented here.

The assumption about the physically reasonable range of the mode width of SA is necessary, since with $\sigma$ values larger than 2.0, the corresponding curves would partly overlap with the previously calculated ones in some areas of the two-dimensional field of the lookup table, meaning that some retrievals with particularly high mode width would have at least two possible solutions. However, to limit the solutions to a maximum $\sigma$ of 2.0 is reasonable firstly, because the associated $r_{med}$ for higher values would become very small (around 10 nm and smaller). Secondly, these higher mode width values are very rarely found in other retrieval works, e.g. Bingen et al. (2004) and Nyaku et al. (2020) don't find values exceeding 1.9. In the in situ measurements by Deshler et al. (2003), where a monomodal lognormal size

**distribution was found as the best fit, 7.6 % of the mode width values exceed 2.3, and only 2.97 % exceed 2.5 (data not**
**shown). For the SAGE III/ISS data set analyzed so far, only very few measurements fall into the space where overlap**
**exists, if for example 2.5 was deemed the maximum realistic value. If for example 2.3 was set as the maximum, even less**
**values would be affected, and so on.**

In an effort to minimize the influence of clouds on the retrieval, all measurements below a tangent height of 25 km at which both the extinction coefficient of the 1021 nm channel is higher than $10^{-4}$ km and the extinction ratio between 449 nm and 1021 nm is lower than 2, suggesting large particles, are excluded from the analysis. This is a rough filter and may exclude some non cloud data, but will generally improve the overall quality of the remaining data.

Once $r_{med}$ and $\sigma$ were determined, the aerosol number density $N$ can easily calculated from the measured extinction coefficient at a single wavelength using the following relation:

$$N = \frac{k_{ext}(\lambda)}{\sigma_{ext}(\lambda)} \tag{4}$$

with $\sigma_{ext}(\lambda)$, the extinction cross section at one of the three wavelengths used for the retrieval, coming from the just retrieved $r_{med}$ and $\sigma$ and $k_{ext}(\lambda)$ being one of the three extinction coefficients from the SAGE III/ISS data set. **In the retrieval data set produced, the wavelength channel at 756 nm was used for these number density calculations, since on average it has the lowest extinction coefficient uncertainties. However, the average difference between the results with this wavelength choice and using one of the other two is below 1%.**

Another useful quantity is the effective radius $r_{eff}$, i.e. the area weighted mean radius (Grainger, 2017). For a lognormal PSD it can be calculated using the following relation:

$$r_{eff} = r_{med} \cdot exp(\frac{5}{2} \cdot ln^2(\sigma)) \tag{5}$$

Two other quantities that can be calculated from the retrieved median radii and mode widths are the mode radius $r_{mod}$ defined as follows:

$$r_{mod} = exp(ln(r_{med}) - ln^2(\sigma)) \tag{6}$$

and $\omega$, a measure of the absolute width of the monomodal lognormal size distribution as introduced by Malinina et al. (2018), which is calculated as follows:

$$\omega = r_{med}^2 \cdot exp(ln^2(\sigma)) \cdot (exp(ln^2(\sigma)) - 1) \tag{7}$$

Both quantities are useful, because they facilitate a more intuitive understanding of changes in the monomodal lognormal size distribution. The mode radius gives the position of the peak of the distribution in linear space. The problem with the mode

width $\sigma$ is, that it is defined relative to the median radius, which means that on its own it does not provide much information on the shape of the size distribution. This is where $\omega$ is useful, since as the standard deviation of the size distribution it is given in absolute units and can therefore be interpreted more easily.

## 4 Error estimation

In order to provide error estimates for the retrieved aerosol size distribution parameters, the sensitivity of the median radius and mode width to the identified main error sources is tested. Those are the quantities, that are the necessary input for the Mie calculations forming the basis of the retrieval method used here, namely the imaginary and real parts of the refractive indices of the $H_2SO_4$-$H_2O$-droplets, as well as the extinction coefficients, which are available in the SAGE III/ISS solar occultation data set. For each of these quantities a separate retrieval with perturbed values is performed to obtain error estimates. The total

errors resulting from that are shown and discussed in chapter 5.

While in the regular retrieval the imaginary part of the refractive index $k$ is treated as being zero for each of the three wavelengths used, for the sensitivity testing values found by Palmer and Williams (1975) are used. While no value was provided for the 449 nm channel, the retrieval here is conducted with $k$ values of $7.6992 \cdot 10^{-8}$ for 756 nm and $1.419 \cdot 10^{-4}$ for the 1544

250 nm channel attained through interpolation of values given in their paper. Since those are still small values no large effect is to be expected.

In the Mie calculations, for each used wavelength a different but fixed value of the real part of the refractive index $n$ is used. Under the assumption of pure sulfate particles, variations in stratospheric temperature and their effect on $n$ are considered to

255 calculate errors, since temperature changes will influence the water vapor pressure in the droplet, changing the composition of the aerosol, when equilibrium is re-established (Steele and Hamill, 1981). Based on the temperature profile data in the SAGE III/ISS level 2 solar occultation data, the atmospheric temperature between 10 km and 30 km altitude, where the aerosol layer is located, and in the time frame from the start of measurements (June 2017) to December 2019 nearly always stayed within $\pm$ 30 K of the 215 K value used in the retrieval. For the sensitivity analysis, the real refractive indices are perturbed according to

260 Lorentz-Lorenz-corrections of the originally used values from 215 K to 245 K. This results in a reduction of $n$ between $0.5\%$ to $0.6\%$, depending on the wavelength. A new retrieval is conducted this way to determine error estimates for the retrieved PSD parameters.

To investigate the effect of the uncertainties of the extinction coefficient values, which are provided in the SAGE III/ISS

data set, those uncertainties are first propagated to get the errors of the extinction ratios forming the basis of the set of curves used for the size parameter retrieval. The consequential error bars, **which** are visualized in Figure 4, can be seen as forming the major and minor axis of an error ellipse around the point, which is used for the regular retrieval. A size distribution parameter retrieval is performed for 8 characteristic points on the error ellipse and the mean value of the resulting median radius and

mode width anomalies is used as the error estimate. Errors using the maximum anomaly obtained this way are also calculated. They are larger, but not discussed here, since they likely overestimate the actual uncertainties of the retrieved values. There is a portion of the used measurements, for which not all of the 8 characteristic points on the error ellipse lie within the confines of the calculated set of curves seen in figure 4, which calls the validity of the respective mean error into question. At 20 km altitude roughly 14 % of the measurements are affected by this. This is why in plots that show total errors of the median radius or mode width, like Figure 8 and 7, these values are excluded in order to not falsify the errors shown.

The total error $\Delta$ is then calculated in the following way, where $\delta_1$, $\delta_2$ and $\delta_3$ are the individual errors resulting from perturbed imaginary and real parts of the refractive indices and propagated extinction coefficient uncertainties, that were just discussed:

$$\Delta = \sqrt{\delta_1^2 + \delta_2^2 + \delta_3^2} \tag{8}$$

To further validate the retrieval method, Ångström exponents calculated with SAGE III/ISS aerosol extinction coefficients at 449 nm and 756 nm **were** compared with Ångström exponents calculated with extinction coefficients obtained from Mie calculations using the retrieved median radii and mode widths. Both results are supposed to be reasonably close to each other. The Ångström exponent is defined as follows:

$$\alpha = -\frac{ln(\frac{k_{ext,449}}{k_{ext,756}})}{ln(\frac{\lambda_{449}}{\lambda_{756}})} \tag{9}$$

with $k_{ext}$ being the extinction coefficient at a specific wavelength and $\lambda$ being that wavelength.

## 5   Results and discussion

At the time of writing, the SAGE III/ISS solar occultation measurements are ongoing. Starting in June 2017 up to around 900 vertical profiles of extinction coefficients for each spectral channel are available per month, from which profiles of the median radius $r_{med}$ and mode width $\sigma$ have been retrieved with the method presented above. Subsequently, the same number of profiles of the mode radius $r_{mod}$, absolute mode width $\omega$, number density $N$ and effective radius have also been calculated.

All values below 25 km, that fall under both criteria of the cloud filtering explained in chapter 3 are excluded. For the whole data set used from June 2017 to December 2019 this is 1.37 % of the retrieved data in general, but only 0.05 % of the retrieved data in 20 km altitude. Of the remaining data all values with an associated accuracy parameter $a$ (see chapter 3) below 16 are excluded, ensuring that particularly noisy data points are filtered out. This removes 8.74 % of the remaining data in general, but only 0.88 % of the data in 20 km altitude.

Figure 6 shows contour plots of averaged vertical profiles from June, 7th to June, 30th 2017 for each of the six quantities listed above. In addition, the extinction coefficient at 449 nm, which is provided directly in the SAGE III/ISS solar occultation data set (NASA, 2019), and the extinction ratio at 449 nm and 756 nm are shown. The contour plots consist of temporal averages of individual profiles sorted in latitude bins of 5°. The red line marks the altitude of the tropopause layer as it is provided in the SAGE III/ISS solar occultation data set.

The month of June 2017 is chosen here because it shows the closest to background conditions of the stratospheric aerosol layer, unperturbed by recent volcanic eruptions or large biomass burning, that is observable in the time frame covered by the SAGE III/ISS measurements at the time of writing. It is important to note, that because of the slow latitude shift of the solar occultation measurements (see Figure 1) due to the orbit of the ISS, there is a time shift in the contour plots shown in Figure 6, i.e. the individual profiles correspond to different dates in the month of June 2017. This latitudinal shift together with the low amount of solar occultation measurements that can be performed by SAGE III/ISS per day (around 30) also means, that showing a higher temporal resolution would not give a better overview. This limits the possibility to observe temporal changes of the retrieved parameters over short time frames, e.g. shorter than a month.

The upper and lower boundaries of the colorbars in Figure 6 roughly mark the ranges within which the values of the respective quantities fall for the data set between June 2017 and December 2019. The median values for this time frame at 20 km altitude are 130.6 nm for the median radius, 1.54 for the mode width $\sigma$, 108.6 nm for the mode radius, 62.4 nm for the absolute mode width $\omega$, 3.17 cm$^{-3}$ for the number density and 188.6 nm for the effective radius. The mode radius is always smaller and the effective radius is always greater than the median radius. These results generally match stratospheric aerosol size distribution parameter retrieval results from other published works in magnitude. However, there is a limited amount of such work in literature, which is not always directly comparable, since different quantities are shown, which are often retrieved at different times and latitudes. For example Bingen et al. (2004) and Fussen et al. (2001) find mode radii roughly between 200 nm and 600 nm in the aftermath of the Mt. Pinatubo eruption in 1991, while in other works the mode radius generally stays smaller than 130 nm (Mclinden et al., 1999; Bourassa et al., 2008; Malinina et al., 2018). Median radii retrieved by Bourassa et al. (2008) lie roughly between 30 nm and 130 nm. The assumed mode widths $\sigma$ can range from monodispersed or effectively 1.0 (Thomason et al., 2008) to 1.4 (Ugolnikov and Maslov, 2018) and 1.6 (Bourassa et al., 2008; Malinina et al., 2018).

Also stratospheric aerosol size distribution parameter retrievals from occultation measurements may lead to systematically larger particle sizes than retrievals from other optical methods, like lidar backscatter measurements, due to differences in the sensitivity of the measurement techniques to larger and smaller aerosols due to different scattering angles (von Savigny and Hoffmann, 2020). This may also in part explain differences between retrievals from occultation and limb measurements. In general, as can be seen in Figure 3, the extinction efficiencies of Mie scattering particles at the wavelengths used are very low for radii lower than roughly 50 nm, which is why most optical measurements will struggle to obtain usable information about particles in that size range.

**It should be noted that one of the major advantages of the retrieval method presented here and using three wavelengths instead of two, is that the lookup table provides unique solutions for almost all of the extinction spectra that can plausibly be observed based on the assumptions made about the aerosol composition and its PSD, i.e. there is exactly**

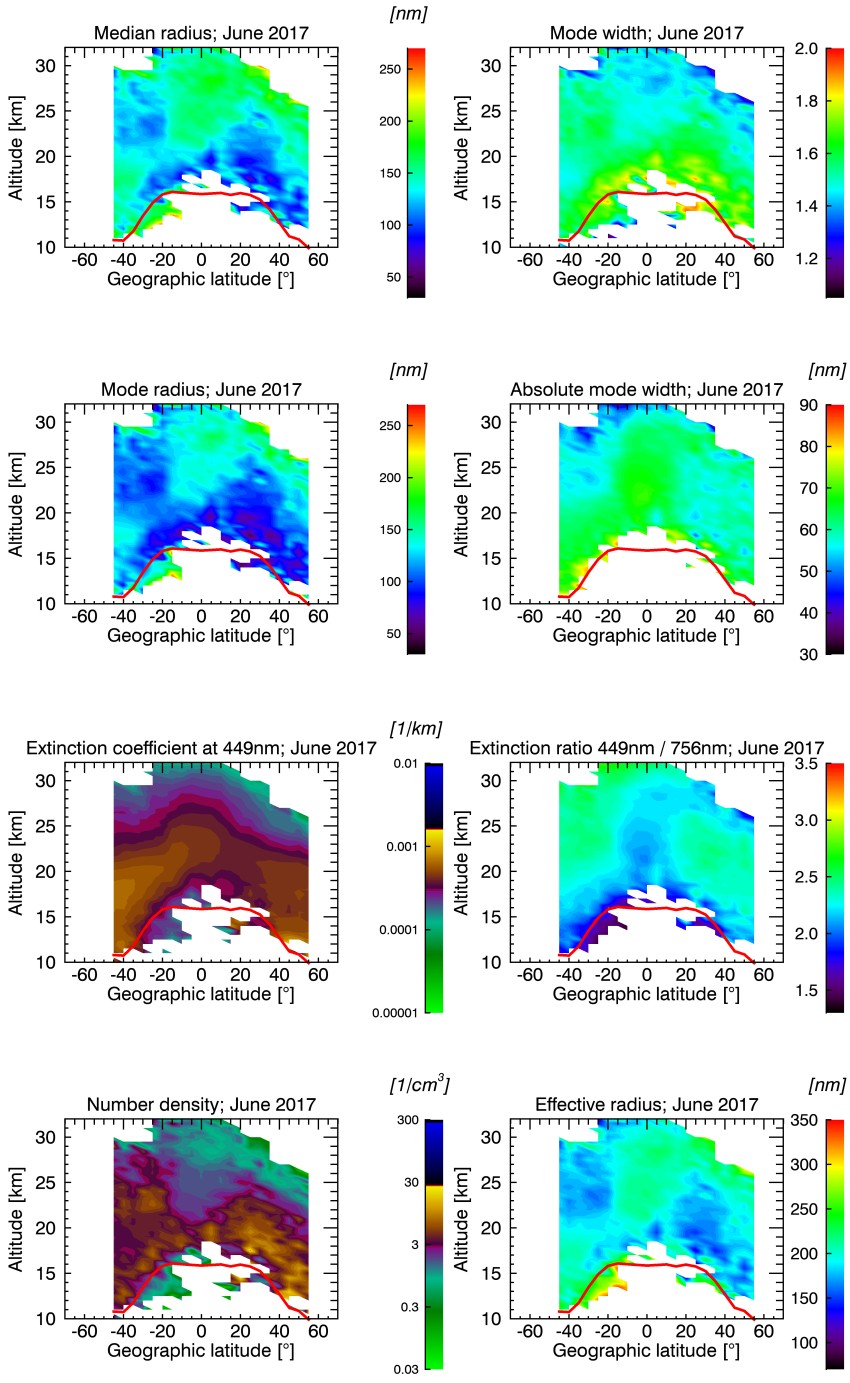

**Figure 6.** Monthly means of median radius, mode width, mode radius, absolute mode width, extinction coefficient at 449 nm, extinction ratio at 449 nm and 756 nm, number density and effective radius for June 2017 with 5° latitude bins. The red line indicates tropopause height.

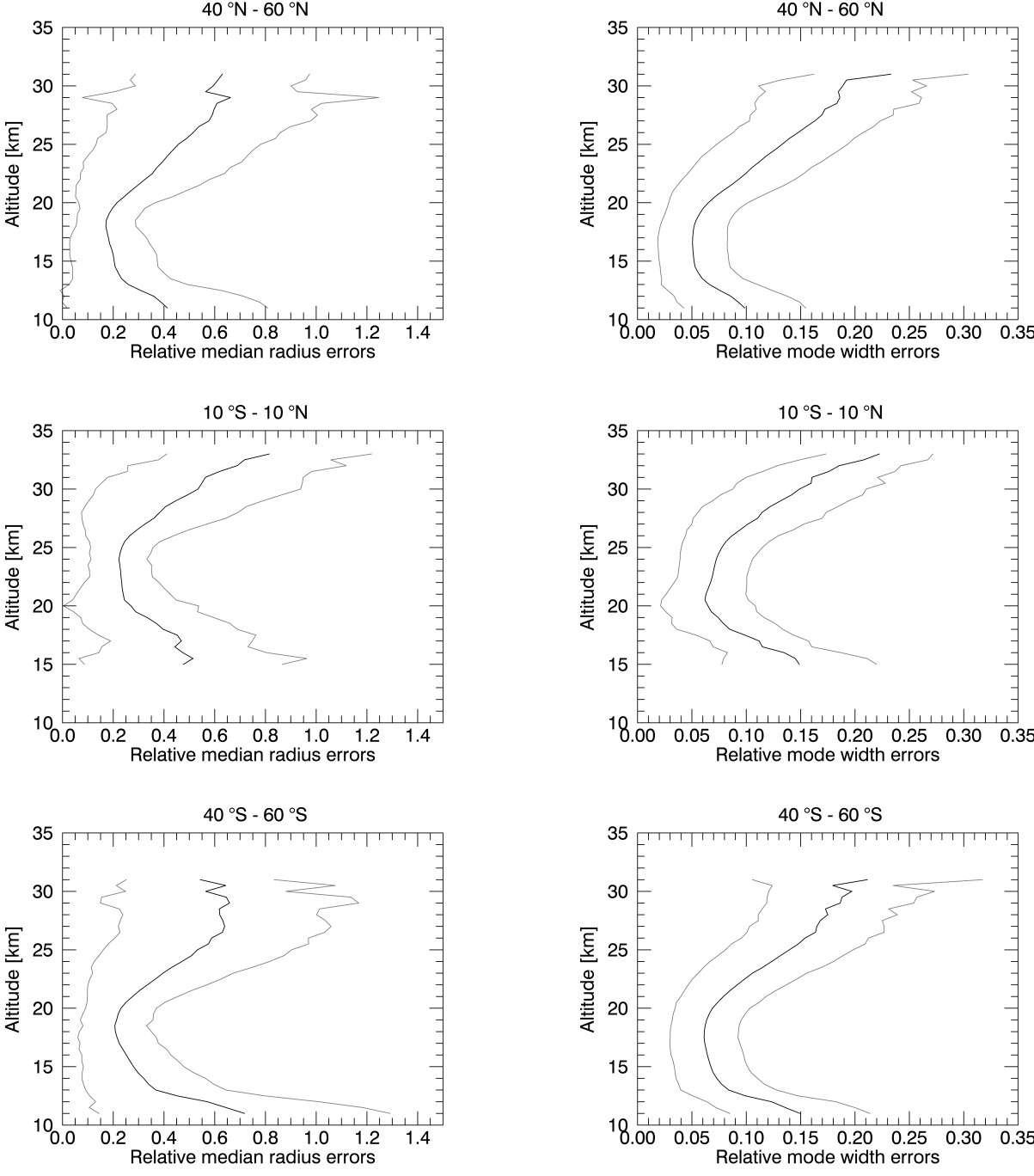

**Figure 7.** Relative total errors of median radius (left column) and mode width (right column) averaged over all measurements between June 2017 and December 2019 shown as black line. Averages are shown between $40°$ N and $60°$ N (upper row), $10°$ S and $10°$ N (middle row) and $40°$ S and $60°$ S (lower row). Grey lines indicate the standard deviation.

**one combination of median radius and mode width that can reproduce the spectral dependence of the extinction put in. This is not the case when using only two spectral channels, since then there is a multitude of possible PSD parameter combinations for a monomodal log-normal distribution that reproduce the same spectral pattern, while having very different median radii, as pointed out by Malinina et al. (2019), making it unclear which one best describes the true conditions in the atmosphere.**

In the results of the current work, median radius and mode width show an anticorrelation at most latitudes and altitudes, which is at least in part because the mode width $\sigma$ is defined relative to $r_{med}$. This means, with an increasing median radius a fixed absolute mode width $\omega$ corresponds to a decreasing $\sigma$. Also an anticorrelation between the extinction ratio at 449 nm and 756 nm and the effective radius as well as the median radius and mode radius can be observed. This is to be expected, since in the wavelength range used in this work, smaller particles have larger Ångström exponents than larger particles, resulting in smaller ratios of extinction coefficients at a smaller wavelength to a larger wavelength. Every contour plot shows a feature in the tropics roughly between 20 km and 25 km, which could be remnants of the eruptions of the Calbuco volcano **(41.3° S, 72.6° W)** on 22nd and 23rd April of 2015, which produced plumes reaching into the stratosphere (Romero et al., 2016). As is to be expected the observable part of the Junge layer shifts in altitude depending on latitude, being lower at higher latitudes.

In Figure 7 relative total errors of the median radius (left column) and mode width (right column) are shown. The total errors were calculated as explained in Sect. 4. The plots show temporal averages from June 2017 to December 2019 in different latitude bins. The top row shows averages between 40° N and 60° N, the middle row shows tropical averages between 10° S and 10° N and the lower row depicts averages between 40° S and 60° S. The respective standard deviations are represented by grey lines. To have representative errors for the retrieval data produced, here also only values with corresponding accuracy parameters above 16 are considered.

In the peak area of the Junge Layer the relative total errors lie between 20 % and 25 % for the median radius and 5 % and 7 % for the mode width. At lower and higher altitudes, where the aerosols contribute less to the overall extinction of solar radiation and fewer measurements deliver analyzable data, the errors as well as their standard deviations become larger. **A perturbation of the real refractive index by 0.5 % to 0.6 % (see Sect. 4), depending on wavelength, resulted in relative uncertainties roughly between 3.5 % and 5 % for the median radius and 0.5 % to 1.3 % for the mode width. A perturbation of the imaginary part of the refractive index as discussed in Sect. 4 resulted in relative uncertainties of roughly 0.4 % to 1 % for $r_{med}$ and 0.15 % to 0.35 % for $\sigma$. These uncertainties are included in the relative total errors, which were described beforehand and are depicted in Figure 7.**

In addition to the contour plots shown before, in Figure 8 two exemplary vertical profiles of the median radius (left panel) and the mode width (right panel) are shown for easier reference of values. Just as in figure 6, the profiles contain averaged values for the month of June 2017. The latitude bin used here contains measurements between 2.5° S and 2.5° N and the grey lines show the total errors, which were calculated as explained in Sect. 4. Here, only measurements with error ellipses lying completely within the set of curves used for the retrieval were used, as was also explained in Sect. 4.

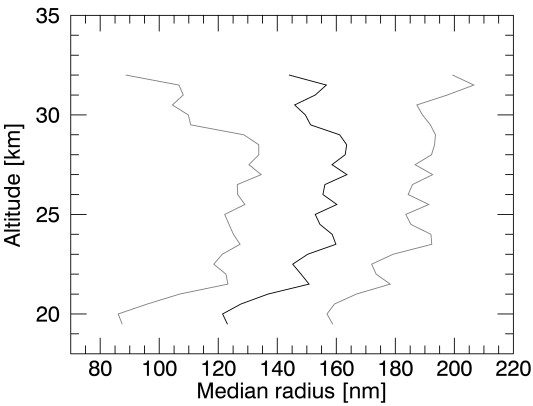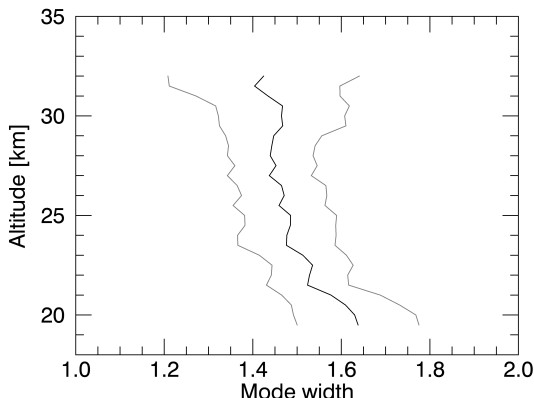

**Figure 8.** Profiles of median radius (left) and mode width (right) averaged between 2.5° S and 2.5° N for the month of June 2017. Grey lines depict total errors as calculated in Sect. 4.

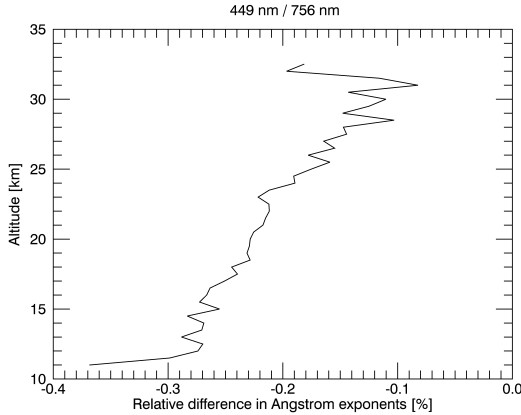

**Figure 9.** Temporal averages of relative differences between Ångström exponents at 449 nm and 756 nm from SAGE III/ISS and from Mie calculations with retrieved PSD parameters from June 2017 to December 2019 in percent.

Additionally, in Figure 9 averaged relative differences between Ångström exponents at 449 nm and 756 nm calculated from the extinction coefficients from SAGE III/ISS and from extinction coefficients obtained with Mie calculations using the median radii and mode widths retrieved with the presented method are shown. This is a measure of the accuracy with which the retrieval algorithm assigns median radius and mode width values to the data points resulting from the extinction ratios of 370 the SAGE III/ISS measurements corresponding to the position within the set of curves (see Figure 4), as well as to what extent the Ångström law correctly describes the spectral dependence of the aerosol extinction. Please note, that the uncertainties of the extinction measurements and therefore also the error bars play no role here. An even distribution of 5 % of the data

between June 2017 and December 2019 is used for this calculation. The relative differences lie between -0.4 % and -0.08 %, which is very small and indicates an accurately working assignment of values in the mentioned step of the retrieval. For the combinations of 756 nm and 1544 nm as well as 449 nm and 1544 nm the differences are even slightly smaller.

## 6   Conclusions

In this work, a novel method to determine size distribution parameters of stratospheric sulfate aerosols based on the SAGE III/ISS solar occultation measurements was implemented. The main purpose of the study is to demostrate this retrieval technique. An analysis of the SAGE III/ISS data set will be presented in a future study.

Due to the wide spectral range covered by the SAGE III/ISS measuring instrument, median radius $r_{med}$ and mode width $\sigma$ of the assumed monomodal lognormal distribution can be retrieved independently, without having to assume one of them beforehand. **Also, using three wavelengths gives unique solutions for the PSD parameters, i.e. unique size distributions, for most atmospheric conditions that can plausibly be expected, which is not the case when using only two spectral channels. Both points are major advantages of this retrieval method over others.** In addition, using occultation measurements has the advantage, that the extinction coefficients were obtained without the need of a priori assumptions **on the aerosols**. However, as discussed, satellite **solar** occultation measurements also come with the limitation of limited spatial and temporal coverage. In addition to $r_{med}$ and $\sigma$, the number density, effective radius, mode radius as well as the absolute mode width has been calculated.

The aerosol particle size retrieval results **are, of course, based on the assumptions about the PSD, but also dependent** on the assumption of the particle composition, since real and imaginary refractive indices are needed for the Mie calculations. Here, the stratospheric aerosol are assumed to be pure $H_2SO_4$-$H_2O$-droplets, which may not be true at all times, e.g. after large biomass burning events (Murphy et al., 2007).

At the peak of the Junge Layer, typical errors lie between 20 % and 25 % for the median radius and 5 % and 7 % for the mode width and increase at higher and lower altitudes. These errors are reasonable, especially since the real errors are very likely smaller, since the extinction coefficients of the different spectral channels of SAGE III/ISS are not completely independent, as it was assumed in this work. Also a comparison between Ångström exponents calculated from the SAGE III/ISS aerosol extinction coefficients and from extinction coefficients calculated with a Mie Code using the retrieved size distribution parameters shows differences smaller than 0.5 %. **Both this Ångström exponent comparison and the low to moderate errors of $r_{med}$ and $\sigma$ suggest,** that this retrieval technique is a solid tool to retrieve aerosol size distribution information from the SAGE III/ISS solar occultation measurements. The data produced in this way can be valuable for comparisons between measurement retrievals and model calculations as well as for the investigation of the impact of volcanic eruptions on climate and atmospheric chemistry.

*Data availability.* The data published in this manuscript can be obtained upon request to the first author. The SAGE III/ISS data was obtained from the NASA Langley Research Center EOSDIS Distributed Archive Center ($https://eosweb.larc.nasa.gov/project/sageiii-iss/sageiii-iss$).

*Author contributions.* CvS initiated the project. FW implemented and further developed the method with assistance by CvS and JZ. LWT provided insights into the SAGE III/ISS instrument and issues related to its measurements. All authors discussed, edited and proofread the manuscript.

*Competing interests.* The authors declare that they have no conflict of interest.

*Acknowledgements.* This work was funded by the Deutsche Forschungsgemeinschaft (DFG, project VolARC (no. 398006378) of the DFG research unit VolImpact (FOR 2820)). We also acknowledge support by the University of Greifswald and thank the Earth Observation Data Group at the University of Oxford for providing the IDL Mie routines used in this study. We also want to thank Elizaveta Malinina for helpful discussions.

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
