# Peer review of "Retrieval of stratospheric aerosol size distribution parameters using SAGE-III/ISS extinction measurements at three wavelengths"

_Atmospheric Measurement Techniques, 2020_

## Referee Comment (RC1) · Anonymous Referee #1 · 14 Oct 2020

General Comments

This paper describes an approach to retrieve the median radius and mode width of a particle size distribution supposed to be lognormal. The method is based on the use of a look-up-table of extinction ratios computed from extinction channels chosen adequately. The method is applied on measurements from the SAGE III experiment on the International Space Station (ISS). This study many similarities with the approach used by Echle et al. (J. Geophys. Res., 103, 19193, 1998), and it might be useful to cite this work which addressed aspects and issues relevant in the present context. The paper is written in a clear way and overall, the methodology and data analysis are

carried out with the necessary care. The topic of this study is very relevant, but some important imprecisions should be first addressed and clarified before publication.

Specific comments

L. 20, p.1-L. 32, p.2: For the completeness, the growing importance of large fires in the feeding of the stratospheric aerosol layer should also be mentioned.

L. 30, p. 2: "while sedimentation . . . aerosol layer": this sentence should be rephrased: sedimentation limits the averaged size, not the size of individual particles.

L. 30, p. 2: "Evaporation . . . temperatures": Since the last aforementioned atmospheric layer is the troposphere, the authors should mention again that they are now considering the stratospheric altitudes.

L. 66-68, p.3 : For the sake of completeness, the authors should also cite works by Bauman et al., (e.g. Bauman et al., J. Geophys. Res., 108, D13, 4382, 2003), and Bingen et al. (e.g. Bingen et al., Ann. Geophys., 21, 797-804, 2002).

L. 116, p.5: "As a result": It is the other way around: Making the assumption that aerosol particles are spherical, the Mie theory can be used.

L. 122-123: Is there any compelling reasons to make this hypothesis (e.g. possible convergence of an iterative process to unrealistic solutions with a mode width out of range), or is it just a matter of defining a realistic range for the LUT ?

Figure 2: It is not clear, from the caption, what is the meaning of the numbers annotated in the figures ("50nm", "60nm", etc.). It is also not clear to which point these values refer. The use of arrows could help specifying the link between the values and corresponding dots (if this is the link the authors mean).

L. 54, p. 2 and L. 131, p.6: The acronym "PSD" should be defined in l. 54, p. 2.

L. 133, p. 6: n and k are determined by the assumption made in l. 114-119, p.5. This should be clarified. It might also be useful to mention, here or in the paragraph on l.

154-157, p.7, that these indexes of refractions are wavelength-dependent.

L. 146-153, p. 7: In §2, it is mentioned that the extinction at 1543.92 nm is based on a 30-nm bandwith channel. While the relative uncertainty for this channel is twice the uncertainty at 1021.20 nm (See Table 1), the uncertainty on the wavelength is much higher than in the case of the other channels, including 1021.20 nm for which the spectral resolution is probably about 2 nm (as mentioned in §2). Did the authors perform a sensivity study to evaluate what the impact of these increased uncertainties (on both the extinction and wavelength) is, how it affects the retrieval of the PSD mode parameters, and to which extend the choice of the 1543.92 nm channel is better than the one of 1021.20 nm (if it is).

L. 163-164, p. 7: How does this interpolation occur ? Following the example give in Fig. 3, the solution for each tangent altitude is likely to cover a large range of sigma-values. Did the author perform some regression ?

L. 165-169, p. 7: I don't see why it is necessary to exclude solutions with large values of ïĄş. In all cases, a range of solutions is likely to provide the set of extinction ranges, taking into account the uncertainty of the different extinction channels. Moreover, a large value of the mode width could be useful as an indication that the assumption made on the aerosol composition is not appropriate (e.g. due to the presence of clouds). In the past and in other frameworks, the exclusion of "unrealistic values" led to overlook unexpected but critical physico-chemical processes, as important as the discovery of the ozone hole. This should make the authors cautious while rejecting values.

L. 170, p.7-l. 184, p. 8: This argumentation is not true because it considers the response of a single particle, and not of a population of particles with a possibly large value of mode width. Hence, it does not take into account the fact that the combination of responses from all individual particles with slightly different radii "blurs" the extinction efficiency signal, in particular in the case of thin particles with respect to the wavelength. It this case, the extinction curves may be much less distinct, and the re-

**AMTD**

trieval of the mode parameters, much less reliable. It should be noted that restricting the allowed range of mode width values may alleviate artificially the problem, leading again to overlook possible solutions.

L. 193, p. 9: How do the authors choose the extinction channel and why ? This should be specified. Would it be meaningful to consider all of them to reduce the uncertainty?

L. 210-213, p. 10: Also the uncertainty on the wavelength might play a role, see comment on l. 146-153, p.7.

L. 239-243, p. 11: See comment on l. 165-169, p. 7, and l. 170, p.7-l. 184, p. 8.

L. 304, p. 15: Please indicate the geolocation of Calbuco to ease the analysis of the figure.

L. 324-325, p.16: The comparison does not only depend on the accuracy of the mode parameter retrieval, but also upon the extend to which the extinction spectral dependence for the actual aerosol population is well described by the Angström law.

L. 337-338, p. 17: This statement has to be qualified and reformulated: indeed, no assumption is required to retrieve the aerosol extinction, but conversely, the authors did use an assumption on the particle size (i.e. lognormal function) to derive expressions of the different mode parameters.

L. 339, p.17: The authors should specify they consider solar occultation in the present case.

L. 346-348, p.17: I am not sure I understand this statement: if the measurements are not independent, off-diagonal terms of the covariance matrix have to be additionally taken into account, and the risk of systematic error may be higher.

Technical corrections

L. 250, p.11: "were compared". L. 261, p.12: Did the authors check that the excluded values are not likely to be due to high aerosol load after a volcanic eruption (e.g., from

their

---

## Referee Comment (RC2) · Anonymous Referee #2 · 2 Nov 2020

Review of AMT-2020-277: Wrana et al., "Retrieval of stratospheric aerosol size distribution parameters using SAGE-III/ISS extinction measurements at three wavelengths"

Summary: Variability of the natural stratospheric aerosol (SA) layer properties relevant to climate and chemistry remains an important field of active research. Wrana et al., present an approach to remotely monitor two properties of the SA particle size distribution using solar occultation measurements like those furnished by the SAGE III/ISS mission. The article reads well, clearly presents the problem and their approach to a solution. It certainly is among the first to apply such an approach to the new SAGE III/ISS data set. However, it is not clear in what way this work is different in princi-

ple from previous publications that have retrieved mono-modal lognormal size distribution properties from multi-wavelength aerosol extinction coefficient measurements , such as Wang et al., 1994 (doi: 10.1029/JD094iD06p08435) or Bingen et al.. 2004 (doi:10.1029/2003JD003518). The article is worthy of publication once the truly 'novel' portions are clearly defined and substantiated.

Comments:

1. Article should include reference to Wang et al., (doi: 10.1029/JD094iD06p08435) who used multi-wavelength SAGE II aerosol extinction to retrieve SA parameters using single-mode lognormal & modified gamma representations.

2. Abstract, first sentence: It is not clear to me what is 'novel' about this approach in view of previously published work.

3. The assumption of composition is understandable in view of the stated research goal to support the investigation of the impact of volcanic eruptions on climate and atmospheric chemistry. However, analysis should be done regarding errors in composition, specifically biomass burning events that have occurred during the first three years of SAGE III/ISS operations. It would be interesting to see when the 'validity-check' with the Angstrom exponent fails. Maybe it is an indicator of a situation of improper composition assumptions.

4. Given that the focus is volcanic eruptions, the authors should examine the case of bimodal size distributions or cases that are more representative of time following an eruption. The conditions of June 2017 were fairly unperturbed with respect to stratospheric aerosols.

5. The authors have a sound approach to choosing wavelengths for the retrieval, paying attention to the quality of the SAGE III/ISS data. However, the relative uncertainties shown in Table 1 are twice as large at 1543nm compared to those at 1021nm. There should be a discussion of how the 'increased information content' available at 1543nm

vs. 1021nm out-weighs the increased uncertainty.

6. Line 148: "...while avoiding potential problems..."

7. In the left panel of Fig. 2 and Fig. 3, how do you reconcile the multiple solutions near the coordinate (0.13, 2)? The narrowest distribution oscillates across several slightly wider distributions.

8. Lines 165-169 mention limiting the width to sigma < 2 to cover cases in general and cites previous work showing values not exceeding 1.9. However, given the limitations of the Mie kernels, it is not clear how the previous work would not have a similar limitation as this current work. That is, does the previous work invoke a similar assumption/limitation on the distribution width?

9. Line 235: "which" instead of "wich"

10. Line 338: "aerosol" instead of "aerol"

11. Line 350 mentions "both validation methods suggest", but it is not clear to me what method other than the Angstrom exponent computation is used for 'validation.'

---

## Author Comment (AC1) · 2 Dec 2020

Comment: This paper describes an approach to retrieve the median radius and mode width of a particle size distribution supposed to be lognormal. The method is based on the use of a look-up-table of extinction ratios computed from extinction channels chosen adequately. The method is applied on measurements from the SAGE III experiment on the International Space Station (ISS). This study many similarities with the approach used by Echle et al. (J. Geophys. Res., 103, 19193, 1998), and it might be useful to cite this work which addressed aspects and issues relevant in the present context. The paper is written in a clear way and overall, the methodology and data

analysis are carried out with the necessary care. The topic of this study is very relevant, but some important imprecisions should be first addressed and clarified before publication.

Reply: We thank the reviewer for his/her constructive, thoughtful and very helpful comments. We tried to answer every question in an appropriate way and implemented almost all of your suggestions. To illustrate some points we attached some figures, which below we will simply call "Figure 1" etc., while if we talk about the figures in the manuscript we will point that out specifically.

Comment: L. 20, p.1-L. 32, p.2: For the completeness, the growing importance of large fires in the feeding of the stratospheric aerosol layer should also be mentioned.

Reply: Thanks for the suggestion, we included it.

Comment: L. 30, p. 2: "while sedimentation ... aerosol layer": this sentence should be rephrased: sedimentation limits the averaged size, not the size of individual particles.

Reply: This sentence has been changed. It now says: "The formed aerosols grow through coagulation and further condensation, while sedimentation limits the averaged aerosol size in the aerosol layer."

Comment: L. 30, p. 2: "Evaporation ... temperatures": Since the last aforementioned atmospheric layer is the troposphere, the authors should mention again that they are now considering the stratospheric altitudes.

Reply: We changed it to: "In the stratosphere, evaporation due to rising temperatures with height generally determines the upper boundary of the aerosol layer"

Comment: L. 66-68, p.3 : For the sake of completeness, the authors should also cite works by Bauman et al., (e.g. Bauman et al., J. Geophys. Res., 108, D13, 4382, 2003), and Bingen et al. (e.g. Bingen et al., Ann. Geophys., 21, 797-804, 2002).

Reply: Thank you for these suggestions. We included citations of both works.

**AMTD**
Comment: L. 116, p.5: "As a result": It is the other way around: Making the assumption that aerosol particles are spherical, the Mie theory can be used.

Reply: Yes, you are right, our wording was unclear here. We reformulated it in an unambiguous and accurate way.

Comment: L. 122-123: Is there any compelling reasons to make this hypothesis (e.g. possible convergence of an iterative process to unrealistic solutions with a mode width out of range), or is it just a matter of defining a realistic range for the LUT ?

Reply: Thank you for your question. Yes, it is in part a matter of defining a physically realistic range for the values covered by the lookup table (LUT). But mainly it is a necessity (as briefly discussed in L. 165-166) due to the behavior of the dependence of the extinction coefficients from Mie calculations on the width of the size distribution. This is illustrated in Figure 1 of the attachment of this reply. In this figure a section of the set of curves used as a LUT in the retrieval is shown. Here, it is extended by curves for larger mode width values, namely 2.1, 2.2, 2.3, 2.5, 2.7 and 3.0 to enable a more intuitive understanding of the problem mentioned in the paper. With sigma values larger than 2.0 there is an overlap of the extinction ratio curves in some areas of the field. In principle this means for the retrieval that there are two possible solutions for the median radius and mode width in these places. This is unfortunate and a limitation of the retrieval method. Firstly however, as can also be seen in Figure 1, these higher mode width values are associated with unrealistically small median radii (around 10 nm and smaller). Secondly such high mode width values are very rarely found in other works. For example, in Figure 2 a histogram of the mode width values for all in situ measurements available from the Wyoming optical particle counter (OPC) measurements, in which a monomodal log normal size distribution was found as the best fit is shown. In this data set 7.6% of the mode width values exceed 2.3, and only 2.97% exceed 2.5. This is not nothing, and has to be kept in mind when analyzing the SAGE III/ISS data set, but in our opinion does not call into question the validity of the data set generated in general. Thirdly, for the SAGE III/ISS data set analyzed so far, only
very few measurements fall into the space where overlap exists, if for example 2.5 was deemed the maximum realistic value, much less so for 2.3, which is illustrated by Figures 3, 4, 5 and 6. In each of these exemplary figures all aerosol extinction ratios from the measurements of SAGE III/ISS from one sunrise or sunset event are shown. Each black dot with error bars represents the measurements at the three wavelengths at one specific tangent altitude. We made no specific selection of measurement events, and are showing the first event as well as the 500th, the 1000th and the 1000th event observed by the instrument.

Of course , the results produced by the retrieval method described in this manuscript have to be interpreted in the context of the assumptions made, as it is the case with all retrievals. Still this particular assumption had to be made to be able to retrieve the respective data. But you pointed out an important issue and we made it more clear in the manuscript.

As a separate issue, regarding your question in parentheses, we want to point out, that here may be a misunderstanding in regard to how the retrieval of the median radius and mode width values in the context of the lookup-table was carried out. The retrieval process is not iterative. For each measurement taken by SAGE III/ISS for a single altitude the retrieval method produces a median radius and a mode width by interpolation between the values of the lookup-table exactly at the position of the measurement data point. The "error bars" play no role in the retrieval itself, only in the identification and exclusion of noisy data. This means, that almost no matter where inside the field of the LUT the data point lands, the retrieval gives a unique result.

Comment: Figure 2: It is not clear, from the caption, what is the meaning of the numbers annotated in the figures ("50nm", "60nm", etc.). It is also not clear to which point these values refer. The use of arrows could help specifying the link between the values and corresponding dots (if this is the link the authors mean).

Reply: Thank you for pointing this out. We added the following description to the

**AMTD**
caption of the figure: "The dots mark r\_med values with increments of 10 nm. The respective r\_med values of five dots are shown in the color corresponding to their mode width."

Comment: L. 54, p. 2 and L. 131, p.6: The acronym "PSD" should be defined in I. 54, p. 2.

Reply: We implemented it.

Comment: L. 133, p. 6: n and k are determined by the assumption made in l. 114-119, p.5. This should be clarified. It might also be useful to mention, here or in the paragraph on l. 154-157, p.7, that these indexes of refractions are wavelength-dependent.

Reply: Thank you, we pointed that out in L. 133 and also implemented your second point.

Comment: L. 146-153, p. 7: In §2, it is mentioned that the extinction at 1543.92 nm is based on a 30-nm bandwith channel. While the relative uncertainty for this channel is twice the uncertainty at 1021.20 nm (See Table 1), the uncertainty on the wavelength is much higher than in the case of the other channels, including 1021.20 nm for which the spectral resolution is probably about 2 nm (as mentioned in §2). Did the authors perform a sensivity study to evaluate what the impact of these increased uncertainties (on both the extinction and wavelength) is, how it affects the retrieval of the PSD mode parameters, and to which extend the choice of the 1543.92 nm channel is better than the one of 1021.20 nm (if it is).

Reply: Yes, the averaged uncertainties of the extinction coefficients of the 1543.92 nm channel are higher than of the 1021.2 nm channel. However, the "distance" between the individual curves (with a specific mode width value) of the lookup-table, which can be seen in the left panel of Figure 2 in the manuscript, is larger for the 1543.92 nm channel. Only together this distance between the curves and the measurement uncertainty (represented by the error bars in Figure 3 in the manuscript) determine how

AMTD
precise the retrieval is, or how big the error of the retrieved parameters is. This is why the "accuracy parameter" which we defined in L. 180 was introduced, which takes account of both factors. In Figure 7 in the attachment of this reply we averaged this accuracy parameter at each altitude of the SAGE III/ISS solar occultation data set over 3000 sunrise and sunset events while using the 1021.2 nm channel (blue line) or the 1543.92 nm channel (red line). As the figure shows, despite larger extinction coefficient measurement uncertainties, the 1543.92 nm channel is suited much better for the retrieval because of how far apart the curves of the LUT are.

Regarding your second point on the uncertainty of the wavelength (if we understand your comment correctly): The uncertainty in the location of the central wavelength is very small compared to the width of the channel. We compared the extinction coefficient from Mie calculations for a fixed assumed monomodal size distribution (median radius=130 nm, sigma=1.6, number density=1 cm^-3) when assuming a monochromatic window exactly at 1543.92 nm with the extinction coefficient resulting from a mean value of each extinction coefficient if the Mie calculation is carried out at each wavelength across the 30 nm wide wavelength window in 0.5 nm steps. The resulting difference between both extinction coefficients was 0.015%, which supports our statement.

Comment: L. 163-164, p. 7: How does this interpolation occur? Following the example give in Fig. 3, the solution for each tangent altitude is likely to cover a large range of sigma-values. Did the author perform some regression?

Reply: See last part of our reply to comment on L. 122-123. The interpolation was performed only for each single data point, the "error bars" play no role in the retrieval itself. Therefore, there is only one solution at each altitude.

Comment: L. 165-169, p. 7: I don't see why it is necessary to exclude solutions with large values of ï ËŻAÂÿs. In all cases, a range of solutions is likely to provide the set of extinction ranges, taking into account the uncertainty of the different extinction
channels. Moreover, a large value of the mode width could be useful as an indication that the assumption made on the aerosol composition is not appropriate (e.g. due to the presence of clouds). In the past and in other frameworks, the exclusion of "unrealistic values" led to overlook unexpected but critical physico-chemical processes, as important as the discovery of the ozone hole. This should make the authors cautious while rejecting values.

Reply: We discuss this above in our reply to the comment on L.122-123. You raise a very good point. We are aware that values should not be excluded without good reason. It is true that not every possible combination of size distribution parameters can be found with our retrieval method. The points we make in our reply to the comment on L.122-123 hopefully explain why this assumption had to be made anyway and that it works very well at least for the conditions that can be found in the SAGE III/ISS data set so far. Each retrieval data set has to be interpreted in the context of its assumptions and this is also the case here.

Comment: L. 170, p.7-l. 184, p. 8: This argumentation is not true because it considers the response of a single particle, and not of a population of particles with a possibly large value of mode width. Hence, it does not take into account the fact that the combination of responses from all individual particles with slightly different radii "blurs" the extinction efficiency signal, in particular in the case of thin particles with respect to the wavelength. It this case, the extinction curves may be much less distinct, and the retrieval of the mode parameters, much less reliable. It should be noted that restricting the allowed range of mode width values may alleviate artificially the problem, leading again to overlook possible solutions.

Reply: We are not completely sure if we understand this comment correctly. Yes, in Figure 4 of the manuscript we only look at single particles to illustrate a general point about the impact of aerosols of different size on the extinction efficiency at different wavelengths, which is part of the calculation of the extinction coefficients for the LUT, as we discuss in L.136-138. But these lookup-tables themselves are based on extinction
coefficients, which are calculated from size distributions, not single aerosol particles.

We can see that our figure and its corresponding text can be confusing in its current position in the flow of the manuscript and moved it to L.139, directly after equation 2 and also tried to word it more clearly. This way there should be a clearer separation from the lookup-tables, which are based on size distributions instead of single aerosol particles.

Comment: L. 193, p. 9: How do the authors choose the extinction channel and why? This should be specified. Would it be meaningful to consider all of them to reduce the uncertainty?

Reply: The extinction channel at 756 nm was used to retrieve the number densities, because it has the lowest average measurement uncertainties of the three. However, the choice does not matter very much, since the difference between the results using different ones of the three wavelengths is very low, on average considerably below 1%. Thank you for pointing this out, we added this information to the manuscript.

Comment: L. 210-213, p. 10: Also the uncertainty on the wavelength might play a role, see comment on l. 146-153, p.7.

Reply: As discussed in our reply to the comment on L. 146-153 the effect of the uncertainty in the position of the center wavelength of the 1543.92 nm channel on the extinction coefficient is very small.

Comment: L. 239-243, p. 11: See comment on l. 165-169, p. 7, and l. 170, p.7-l. 184, p. 8.

Reply: We are not fully sure, which points regarding the other comments on L.165-169 and L.170-184 this comment is pointing to. Hopefully we have covered the topic in our replies to the respective comments.

Comment: L. 304, p. 15: Please indicate the geolocation of Calbuco to ease the analysis of the figure.
Reply: Thank you for this suggestion. We added the information.

Comment: L. 324-325, p.16: The comparison does not only depend on the accuracy of the mode parameter retrieval, but also upon the extend to which the extinction spectral dependence for the actual aerosol population is well described by the Angström law.

Reply: We included this in our sentence as follows: "This is a measure of the accuracy with which 325 the retrieval algorithm assigns median radius and mode width values to the data points resulting from the extinction ratios of the SAGE III/ISS measurements corresponding to the position within the set of curves (see Figure 3), as well as to what extent the Ångström law correctly describes the spectral dependence of the aerosol extinction."

Comment: L. 337-338, p. 17: This statement has to be qualified and reformulated: indeed, no assumption is required to retrieve the aerosol extinction, but conversely, the authors did use an assumption on the particle size (i.e. lognormal function) to derive expressions of the different mode parameters.

Reply: Perhaps the sentence is misleading. It is certainly correct that our size parameter retrieval requires assumptions (mono-modal log-normal PSD), which are probably quite strong assumptions. However, the retrieval of aerosol extinction coefficients from the solar occultation measurements does not require any a priori assumptions on the PSD. For aerosol retrievals from, e.g., limb-scatter measurements an aerosol PSD has to be assumed for extinction coefficient profile retrievals.

Comment: L. 339, p.17: The authors should specify they consider solar occultation in the present case.

Reply: Thanks for pointing this out. Stellar occultation certainly has a better geographical coverage. We replaced "satellite occultation measurements" by "satellite solar occultation measurements".

Comment: L. 346-348, p.17: I am not sure I understand this statement: if the mea-
surements are not independent, off-diagonal terms of the covariance matrix have to be additionally taken into account, and the risk of systematic error may be higher.

Reply: The reviewer is correct, that the covariance terms have to be taken into account in the Gauss' error propagation law if the extinction measurements at different wavelengths are not independent. However, one of the partial derivatives the covariances are multiplied with, is negative. For that reason the total variance term will decrease if the covariances are considered.

Comment: Technical corrections L. 250, p.11: "were compared". L. 261, p.12: Did the authors check that the excluded values are not likely to be due to high aerosol load after a volcanic eruption (e.g., from their

Reply: Thank you for pointing out the typing error. Regarding the second part of the comment (a part of this comment unfortunately seems to be missing): Yes, we looked at the data that is removed by this cloud filter. It mostly removed tropospheric data without interfering with the signals which are visible after the Ambae (April and July 2018), Raikoke and Ulawun (both June 2019) eruption.

AMTD

---

## Author Comment (AC2) · 2 Dec 2020

Comment: Summary: Variability of the natural stratospheric aerosol (SA) layer properties relevant to climate and chemistry remains an important field of active research. Wrana et al., present an approach to remotely monitor two properties of the SA particle size distribution using solar occultation measurements like those furnished by the SAGE III/ISS mission. The article reads well, clearly presents the problem and their approach to a solution. It certainly is among the first to apply such an approach to the new SAGE III/ISS data set. However, it is not clear in what way this work is different in principle from previous publications that have retrieved mono-modal lognormal size distribution properties from multi-wavelength aerosol extinction coefficient measurements , such as Wang et al., 1994 (doi: 10.1029/JD094iD06p08435) or Bingen et al., 2004 (doi:10.1029/2003JD003518). The article is worthy of publication once the truly 'novel' portions are clearly defined and substantiated.

Reply: Thank you for your helpful comments. We tried to answer every comment in an appropriate way. Other methods to determine information on stratospheric aerosol PSD were certainly applied in the past (e.g. the papers by Bingen, which are cited in our paper) and they are of course also important. The novel aspect of our approach is: - The use of more than one extinction ratio provides more robust estimates of median radius AND width of the assumed mono-modal log-normal PSD. This is possible due to the broader spectral range of SAGE III/ISS (up to 1544 nm) as opposed to SAGE II (up to 1021 nm). - Retrieving only, e.g. median radius and fixing the width does not allow for a unique solution. As recently pointed out by Malinina et al. (2019), one extinction ratio (or one Angstrom-exponent) may lead to distributions with very different mean radii. This is not the case with the retrieval method presented in the manuscript, where unique solutions are found for almost all SAGE III/ISS measurements, except for some combinations of extinction ratios where 2 solutions are possible if very large mode width values together with particularly small median radius values are allowed.

Comment: 1. Article should include reference to Wang et al., (doi: 10.1029/JD094iD06p08435) who used multi-wavelength SAGE II aerosol extinction to retrieve SA parameters using single-mode lognormal & modified gamma representations.

Reply: Thanks for the suggestion, we included citation of this work.

Comment: 2. Abstract, first sentence: It is not clear to me what is 'novel' about this approach in view of previously published work.

Reply: See our response to the general comment above. We added a brief statement to the abstract and the conclusion to make the novel aspects of the described approach
more evident.

Comment: 3. The assumption of composition is understandable in view of the stated research goal to support the investigation of the impact of volcanic eruptions on climate and atmospheric chemistry. However, analysis should be done regarding errors in composition, specifically biomass burning events that have occurred during the first three years of SAGE III/ISS operations. It would be interesting to see when the 'validity-check' with the Angstrom exponent fails. Maybe it is an indicator of a situation of improper composition assumptions.

Reply: The reviewer raises a very important point here. Indeed, the optical properties, namely the refractive indices, of aerosols coming from biomass burning events will probably differ from the ones we used assuming a composition of only sulfuric acid and water. Sadly, a detailed discussion of this topic would probably go beyond the scope of this study, as in literature a wide range of both the real (roughly between 1.34 and 1.9) and imaginary part ( $\sim 0.0082$  to 0.468) of the refractive indices of aerosol from biomass burning can be found (Bluvshtein et al. , 2017, doi:10.1002/2016JD026230 ; Poudel et al., 2017, doi:10.3390/atmos8110228 ; Sarpong et al., 2020, doi:10.3390/atmos11010062). The composition and therefore the refractive index can vary based on the burning fuel, e.g. the type of wood, and other factors like burning conditions. In addition, values are usually only provided for very specific wavelengths, as opposed to the broad wavelength spectrum we would need for error calculations. Lastly these studies usually provide tropospheric values, which may not be enough to make a statement about stratospheric conditions, since at least mixing with background sulfate aerosols will probably play a role.

However, since we perturbed the refractive indices in our error calculations, we included information on the error of the median radius and mode width that resulted from a perturbation of the refractive indices by a certain value, depending on wavelength, in the manuscript. This can help to at least broadly evaluate the effect on median radius and mode width, should certain values of the complex refractive index be assumed. AMTD
Also, following your suggestion, in Figure 1 in the attachment we show the results of the comparison of the Angstrom exponents in the same way as explained in the manuscript, but averaged for October 2017 and between 30°N and 60°N only. In this time and latitude window the effects of the North American Wildfires of 2017 are clearly visible in the aerosol parameters (such as median radius and effective radius). But the test showed roughly the same result as when we averaged over the whole data set from June 2017 to December 2019 (as in the manuscript). This confirms our thinking, that the test comparing the Angstrom exponents should not be a helpful tool to identify cases where the assumptions about the aerosol composition fails. This is because, it only tests if a comparable spectral dependence of the aerosol extinction can be obtained from the extinction coefficients provided in the SAGE III/ISS data set and from Mie calculations with the size distribution parameters which were retrieved from those extinction coefficients. Even if the refractive indices used for the lookup-table and therefore the retrieval of the particle size distribution parameters would not correctly represent the optical properties of the actual aerosol population, the output of the retrieval would be a size distribution which can reproduce the same extinction ratios or Angstrom exponents, apart from the errors due to the interpolation process itself. This is why we described the test as a validity check for how accurately "[...] the retrieval algorithm assigns median radius and mode width values to the data points resulting from the extinction ratios of the SAGE III/ISS measurements corresponding to the position within the set of curves [...]." in the manuscript.

Comment: 4. Given that the focus is volcanic eruptions, the authors should examine the case of bimodal size distributions or cases that are more representative of time following an eruption. The conditions of June 2017 were fairly unperturbed with respect to stratospheric aerosols.

Reply: Unfortunately, it is not really possible to retrieve parameters of a bi-modal PSD from the SAGE measurements, because the number of free parameters is too large. If two mono-modal log-normal modes are assumed one would need to retrieve median
radius, width and number density for both modes, i.e. 6 parameters. The conditions of June 2017 were chosen on purpose because they best represent background conditions of the months that are available in the SAGE III/ISS data set at this point in time. While this paper is an introduction of the retrieval method itself, in a follow-up paper different volcanic eruptions during the SAGE III/ISS operation and the evolution of the stratospheric aerosol size distribution in their aftermath are supposed to be discussed.

Comment: 5. The authors have a sound approach to choosing wavelengths for the retrieval, paying attention to the quality of the SAGE III/ISS data. However, the relative uncertainties shown in Table 1 are twice as large at 1543nm compared to those at 1021nm. There should be a discussion of how the 'increased information content' available at 1543nm vs. 1021nm out-weighs the increased uncertainty.

Reply: Thank you for this suggestion, we agree and added a discussion about that topic in the manuscript. While the averaged uncertainties of the extinction coefficients of the 1543.92 nm channel are higher than of the 1021.2 nm channel, the "distance" between the individual curves (with a specific mode width value) of the lookup-table, which can be seen in the left panel of Figure 2 in the manuscript, is larger for the 1543.92 nm channel. Only together this distance between the curves and the measurement uncertainty (represented by the error bars in Figure 3 in the manuscript) determine how precise the retrieval is, or how big the error of the retrieved parameters is. This is why the "accuracy parameter" which we defined in L. 180 was introduced, which takes account of both factors. In Figure 2 in the attachment of this reply we averaged this accuracy parameter at each altitude of the SAGE III/ISS solar occultation data set over 3000 sunrise and sunset events while using the 1021.2 nm channel (blue line) or the 1543.92 nm channel (red line). As the figure shows, despite larger extinction coefficient measurement uncertainties, the 1543.92 nm channel is suited much better for the retrieval because of how far apart the curves of the LUT are.

Comment: 6. Line 148: ": : :while avoiding potential problems: : :"
Reply: Included.

Comment: 7. In the left panel of Fig. 2 and Fig. 3, how do you reconcile the multiple solutions near the coordinate (0.13, 2)? The narrowest distribution oscillates across several slightly wider distributions.

Reply: Sadly, there is not much we can do about it. Even though it would be better, if this ambiguity did not exist, the very small percentage of data that falls into this narrow area of multiple solutions can be flagged and excluded from analysis.

Comment: 8. Lines 165-169 mention limiting the width to sigma

works.

Comment: 9. Line 235: "which" instead of "wich"

Reply: Thank you for pointing out the typing error. We fixed it.

Comment: 10. Line 338: "aerosol" instead of "aerol"

Reply: Fixed.

Comment: 11. Line 350 mentions "both validation methods suggest", but it is not clear to me what method other than the Angstrom exponent computation is used for 'validation.

Reply: Yes, you are right, our wording was wrong here. The sentence was referring to the reasonable total errors of the size distribution parameters as well as the comparison of the Angstrom exponents. We changed it accordingly.
**Fig. 1.** Relative difference between Angstrom exp. from SAGE III/ISS and from Mie calculations with retrieval results averaged for october 2017 between 30°N and 60°N (Canadian Wildfires).

---

## Editor Decision (ED1)

AMT_2020_277

The authors have adequately addressed the major points raised by the reviewers. A few minor details, however, must be addressed before the manuscript can be accepted for publication.

L. 50, p. 2: "…the reason Robock…" instead of "…the reason, why Robock…"

L. 72, p. 3: The authors mean "Bingen et al., 2003". Also need to include Wang et al. 1989, as they predate Bingen in retrieving a PSD from occultation data.

L. 138-141, p.7: "the total number density N0 (…) has to be assumed": This is a little bit confusing, because it gives the impression that the authors do not retrieve the total number density, what is wrong. It would be less confusing to write, e.g., that "the single aerosol extinction coefficient is calculated with the Mie Code (Oxford, 2108) for a total number density equal to 1" – I guess this is the case. Please note also that "coefficient" is singular. "Extinction coefficient values" can also be used.

L. 161, p. 8; Figure 4: "at a different tangent height": The authors should mention the range covered by the tangent height values used in Figure 4. It might be useful to use a color code (e.g. grey tones) on the plot for the error bars to visualize in a glance in which altitude range every measurement point is situated.

L. 165, p. 9: "coordinates of the measurement data point": The authors should specify which kind of coordinates they are referring to.

L. 170, p. 9 : "..ratios where sets…" instead of "…ratios, where sets…"

L. 182, p.9: "the precision": It does not determine the precision, but the accuracy.

L. 188-190: "accuracy parameter": The authors should indicate if this accuracy parameter is some standard parameter (e.g. by providing a reference) or a parameter they are defining ("We define the accuracy parameter as …").

L. 199, p.10: "plot", lowercase.

L. 217, p.11: "can easily be calculated".

L. 368-371, p.18: This sentence is particularly long and difficult to read. I suggest the authors to rephrase it.

L. 373, p.19: The authors should specify which "relative difference" they are talking about.

---

## Author Response (AR2)

**Author replies to referee comments – second revision**

**Comment:** L.50, p. 2: "...the reason Robock..." instead of "...the reason, why Robock..."

**Reply:** We changed it as suggested.

**Comment:** L. 72, p. 3: The authors mean "Bingen et al., 2003". Also need to include Wang et al. 1989, as they predate Bingen in retrieving a PSD from occultation data.

**Reply:** Thank you for spotting the error. We also included the citation of Wang et al. 1989 at this point in the manuscript.

**Comment:** L. 138-141, p.7: "the total number density N0 (...) has to be assumed": This is a little bit confusing, because it gives the impression that the authors do not retrieve the total number density, what is wrong. It would be less confusing to write, e.g., that "the single aerosol extinction coefficient is calculated with the Mie Code (Oxford, 2108) for a total number density equal to 1" –I guess this is the case. Please note also that "coefficient" is singular. "Extinction coefficient values" can also be used.

**Reply:** Thanks for pointing this out. In order to make it more clear that in these lines we are talking about the Mie calculations of the lookup-table and not the quantities we are retrieving, we changed the wording in this paragraph in several places, eg. from "To obtain a total aerosol extinction coefficient (...)" to "To obtain a theoretical total aerosol extinction coefficient (...)". Also we changed "single aerosol extinction coefficients" to "single aerosol extinction coefficient values".

However, although it was the case in our calculations, the number densities do not necessarily have to be set to 1 in the Mie calculations for the median radius and mode width retrieval to work. As we pointed out in L. 140-143, this is because the retrieval uses extinction ratios, which are independent of the number density, since the number density cancels out, when a ratio of two extinction coefficients is formed.

**Comment:**L. 161, p. 8; Figure 4: "at a different tangent height": The authors should mention the range covered by the tangent height values used in Figure 4. It might be useful to use a color code (e.g. grey tones) on the plot for the error bars to visualize in a glance in which altitude range every measurement point is situated.

**Reply:** The tangent altitudes of the measurement points shown range from 18.5 km to 32 km for this particular occultation event. We included this information in the manuscript and also followed your suggestion of using different grey tones for different tangent height intervals in Figure 4.

**Comment:**L. 165, p. 9: "coordinates of the measurement data point": The authors should specify which kind of coordinates they are referring to.

**Reply:** We replaced "coordinates of the measurement data point" by "extinction ratio coordinates of the measurement data point within the 2-D space of the lookup table".

**Comment:**L.170, p.9 : "..ratios where sets..." instead of "...ratios, where sets..."

**Reply:** Implemented.

**Comment:**L. 182, p.9: "the precision": It does not determine the precision, but the accuracy.

**Reply:** We changed "precision" to "accuracy".

**Comment:**L. 188-190: "accuracy parameter": The authors should indicate if this accuracy parameter is some standard parameter (e.g. by providing a reference) or a parameter they are defining ("We define the accuracy parameter as …").

**Reply:** Thanks for the suggestion. We included it as follows: "We define an accuracy parameter below, (...)"

**Comment:**L. 199, p.10: "plot", lowercase.

**Reply:** We changed it.

**Comment:**L. 217, p.11: "can easily be calculated".

Reply: Implemented.

**Comment:**L. 368-371, p.18: This sentence is particularly long and difficult to read. I suggest the authors to rephrase it.

**Reply:** We shortened the sentence to "This is a measure of the accuracy with which the retrieval algorithm assigns median radius and mode width values to the measurement data points via interpolation (see Figure 4), as well as to what extent the Angström law correctly describes the spectral dependence of the aerosol extinction."

**Comment:**L. 373, p.19: The authors should specify which "relative difference" they are talking about.

**Reply:** We changed the sentence to "The relative differences between the Angström exponents calculated from the SAGE III/ISS data and the ones calculated with the retrieved size distribution parameters lie between -0.4 % and -0.08 %, which is very small and indicates an accurately working assignment of values in the mentioned step of the retrieval."